green chemistry/organic chemistry

αβ-unsaturated ketones, alkenes, perovskite, oxidative coupling, LaCoO₃, catalysis

**Authors for correspondence:**
Tung T. Nguyen
e-mail: tungtn@hcmut.edu.vn
Nam T. S. Phan
e-mail: ptsnam@hcmut.edu.vn

This article has been edited by the Royal Society of Chemistry, including the commissioning, peer review process and editorial aspects up to the point of acceptance.

# Alternative pathways to α,β-unsaturated ketones via direct oxidative coupling transformation using Sr-doped LaCoO₃ perovskite catalyst

Khang H. Trinh, Son H. Doan, Tien V. Huynh, Phuong H. Tran, Diep N. Pham, Minh-Vien Le, Tung T. Nguyen and Nam T. S. Phan

Faculty of Chemical Engineering, HCMC University of Technology, VNU-HCM, 268 Ly Thuong Kiet, District 10, Ho Chi Minh City, Viet Nam

(iD) M-VL, 0000-0002-6651-0121; TTN, 0000-0002-0912-9981; NTSP, 0000-0002-5928-7638

A strontium-doped lanthanum cobaltite perovskite material was prepared, and used as a recyclable and effective heterogeneous catalyst for the direct oxidative coupling of alkenes with aromatic aldehydes to produce α,β-unsaturated ketones. The reaction afforded high yields in the presence of di-*tert*-butylperoxide as oxidant. Single oxides or salts of strontium, lanthanum and cobalt, and the undoped perovskite offered a lower catalytic activity than the strontium-doped perovskite. Benzaldehyde could be replaced by benzyl alcohol, dibenzyl ether, 2-oxo-2-phenylacetaldehyde, 2-bromoacetophenone or (dimethoxymethyl) benzene in the oxidative coupling reaction with alkenes. To our best knowledge, reactions between these starting materials with alkenes are new and unknown in the literature.

## 1. Introduction

α,β-Unsaturated ketones and their derivatives are precious structures, predominantly distributed in numerous biologically active natural products, medicinal chemicals, agricultural chemicals and functional organic materials [1–3]. These structures also serve as starting materials for the synthesis of many valuable organic chemicals [4–6]. Classically, they were synthesized via condensation reactions in the presence of stoichiometric amounts

of strong bases, thus suffering several severe drawbacks [7]. Accordingly, considerable efforts have been devoted to achieve more efficient and greener synthetic pathways to these significant compounds. Some recent protocols include platinum-catalysed tandem hydroformylation/aldol condensation reaction of vinyl aromatics and ketones [8], Doebner–Knoevenagel type condensation reactions in the presence of acyl/aroyl Meldrum's acid [9], copper-catalysed Saegusa-type oxidation of enol acetates [10], zeolite-catalysed tandem hydration/condensation of alkynes and aldehydes [11] and IBX-catalysed dehydration of allylic hydroperoxides [12]. Among numerous synthetic approaches, reactions via direct C–H bond activation have been explored, as the prefunctionalization of reactants and the purification of intermediates could be minimized [13–15]. Previously, Wang *et al.* developed the homogeneous copper-catalysed direct oxidative coupling of alkenes with aldehydes to produce α,β-unsaturated ketones (scheme 1a) [16]. In this protocol, neither the aldehydes nor the alkenes need to be prefunctionalized, therefore offering advantages over conventional synthetic strategies.

Perovskites have emerged as a significant family of materials with fascinating physico-chemical properties, such as electron mobility, redox performance, thermal stability, and electronic and ionic conductivity [17–19]. They are complex metal oxides with the general formula of $ABO_3$, where A normally stands for an alkaline-earth or rare-earth metal cation, and B represents a transition metal [20,21]. In these structures, partial or full substitution of the A or B cation leads to different interesting properties [22]. In the catalysis field, perovskites have been extensively studied for reduction–oxidation transformations for many years, including CO oxidation [23–26], reforming of methane [27–30], production of hydrogen [31–33], removal of volatile organic compounds [34–36], wastewater treatment [37–40] water splitting [41–44] and oxygen reduction reaction [45–48]. Nevertheless, the utilization of perovskites as catalysts for organic synthesis is *extremely rare* in the literature. In this study, we would like to report the application of a strontium-doped lanthanum cobaltite perovskite as a recyclable heterogeneous catalyst for the direct oxidative coupling of alkenes with aromatic aldehydes to produce α,β-unsaturated ketones (scheme 1b). It should be noted that single oxides or salts of strontium, lanthanum and cobalt, and the undoped perovskite exhibited lower activity than the strontium-doped perovskite. Interestingly, we found that aromatic aldehydes could be replaced by several starting materials in the perovskite-catalysed direct oxidative coupling reaction (scheme 1c–g). To our best knowledge, reactions between these starting materials and alkenes are new and unknown in the literature.

Previous work [16]:

This work:

**Scheme 1.** The difference between previous work and this work.

# 2. Experimental set-up

## 2.1. Synthesis of strontium-doped lanthanum cobaltite perovskite catalyst

The strontium-doped lanthanum cobaltite perovskite powder was prepared via gelation and calcination approach [49]. The nitrate salts of lanthanum ($La(NO_3)_3.6H_2O$, 3.90 g, 0.009 mol), strontium ($Sr(NO_3)_2$, 1.28 g, 0.006 mol) and cobalt ($Co(NO_3)_2.6H_2O$, 4.45 g, 0.015 mol) were dissolved in deionized water (100 ml) with vigorous stirring. Subsequently, citric acid (11.3 g, 0.06 mol) as a chelating agent was added to the solution. The resulting mixture was then heated at 80°C under continuous stirring for 4 h in a water bath, and the clear pink solution was transformed into a red gel. After the gelation step, the red gel was burnt at 140°C overnight in an electric oven to produce soft-ash form. In order to obtain the perovskite, the ash was pre-calcined at 400°C for 0.5 h, pulverized, and then calcined at 500°C under air for 8 h. The resulting powder was consequently ball-milled in ethanol media using 5 mm diameter zirconia balls for 12 h at room temperature, and dried overnight in an electric oven at 140°C to achieve the strontium-doped lanthanum cobaltite perovskite ($La_{0.6}Sr_{0.4}CoO_3$, 2.56 g).

## 2.2. Catalytic studies

In a representative experiment, a mixture of 1,1-diphenylethylene (0.0541 g, 0.3 mmol) and diphenyl ether (16 µl, 0.1 mmol) as an internal standard in benzaldehyde (1 ml) was added into a pressurized vial. The reaction mixture was magnetically stirred and heated for 5 min. The strontium-doped lanthanum cobaltite perovskite catalyst (3.4 mg, 5 mol%) was then added into the vial. The reaction mixture was magnetically stirred for 2 min to disperse entirely the catalyst in the liquid phase, followed by the addition of di-*tert*-butylperoxide (DTBP; 0.27 ml, 1.2 mmol). The resulting mixture was continuously stirred at 120°C for 18 h under air. Upon completion of the reaction step, the reaction mixture was diluted with ethyl acetate (5 ml) and washed with saturated $NaHCO_3$ solution (5 ml). The ethyl acetate layer was dried by anhydrous $Na_2SO_4$. Reaction yields were recorded from GC analysis results regarding the diphenyl ether internal standard. To isolate the coupling product, the ethyl acetate solution was concentrated under vacuum and purified by column chromatography on silica gel with hexane/ethyl acetate solvent mixture. The structure of the coupling product was analysed by GC-MS, [1]H NMR and [13]C NMR. The $La_{0.6}Sr_{0.4}CoO_3$ catalyst was separated from the reaction mixture by centrifugation, washed carefully with ethylacetate, acetone and diethyl ether, dried under vacuum at room temperature overnight on a Schlenkline, and re-used to investigate the possibility of catalyst recycling.

# 3. Results and discussion

## 3.1. Catalyst synthesis and characterization

The strontium-doped lanthanum cobaltite perovskite powder was characterized using conventional analysis methods. Scanning electron microscopy (SEM) studies indicated that particles are of inhomogeneous size, highly agglomerated, forming irregular lumps (figure 1). Energy-dispersive X-ray (EDX) investigation confirmed the presence of La, Sr, Co and O elements in the structure of the perovskite with La : Sr : Co molar ratio of 0.63 : 0.37 : 1.07 being recorded (figure 2). Additionally, ICP analysis of the bulk sample indicated a La : Sr : Co molar ratio of 0.63 : 0.38 : 1.0. These data are in good agreement with the designed ratio of 0.60 : 0.40 : 1.0, confirming the $La_{0.6}Sr_{0.4}CoO_3$ formula for the material. Nitrogen physisorption measurements of the $La_{0.6}Sr_{0.4}CoO_3$ sample revealed Langmuir surface areas of less than $10\ m^2\ g^{-1}$ (figure 3). X-ray diffraction (XRD) patterns (figure 4) showed the presence of the typical $LaCoO_3$ perovskite diffraction peaks at 2θ of 23.3°, 32.9°, 40.7°, 47.4° and 48.8°. However, the XRD patterns also suggested that the material contained impurities phases of SrO (at 2θ of 29.5° and 35.2°) and $La_2O_3$ (at 2θ of 50.0°). The analysis results verified that $La_{0.6}Sr_{0.4}CoO_3$ is well indexed to perovskite structure of $LaCoO_3$ (JCPDS 82–1961). Certainly, XRD result of the $La_{0.6}Sr_{0.4}CoO_3$ is consistent with the literature [49–51]. Fourier-transform infrared spectroscopy (FT-IR) spectra of the $La_{0.6}Sr_{0.4}CoO_3$ in the region between 4000 and $500\ cm^{-1}$ are identical to those previously reported in the literature (figure 5) [52]. The absorption bands at $574\ cm^{-1}$ correspond to the deformation modes of $LaCoO_3$. Another absorption peak at $420\ cm^{-1}$ (not shown here) marks the formation of O–Co–O and La–O–Co, indicating that $LaCoO_3$ structure was generated [53]. The strong absorption peak at $1454\ cm^{-1}$ is due to the vibrations of the lanthanum cobaltite crystal lattice [54].

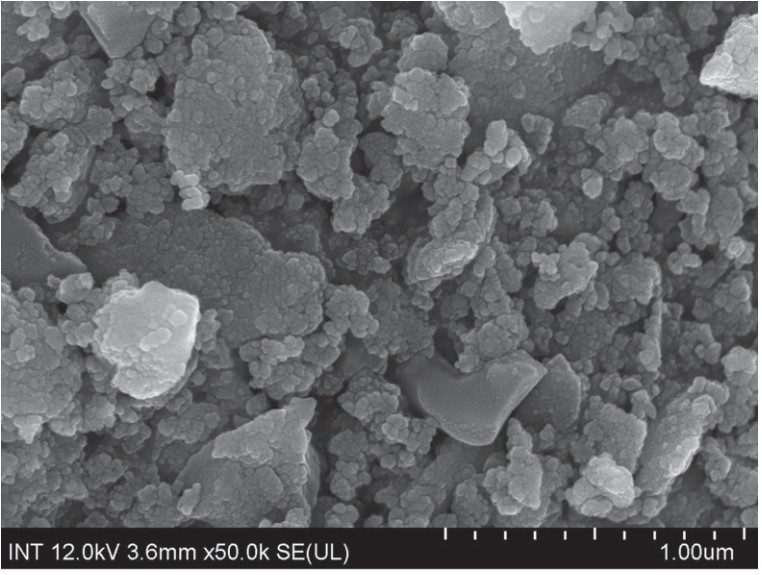

**Figure 1.** SEM micrograpth of the $La_{0.6}Sr_{0.4}CoO_3$ perovskite catalyst.

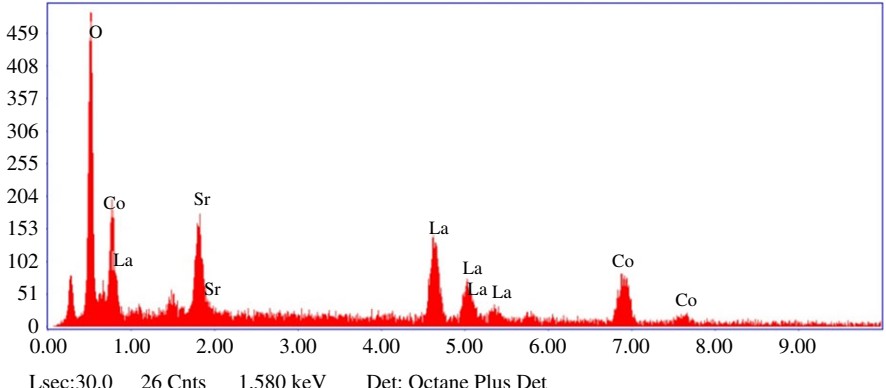

**Figure 2.** EDX patterns of the $La_{0.6}Sr_{0.4}CoO_3$ perovskite catalyst.

A band at 667 cm$^{-1}$ was observed on the sample, assigned to $Co_3O_4$, which was generated due to strontium doping in the perovskite [55]. Thermal gravimetric analysis (TGA) result verified that the material was thermally stable up to 600°C (figure 6). Transmission electron microscopy (TEM) studies showed a regular shape of the material with the nanosize of 5–15 nm (figure 7).

## 3.2. Catalytic studies

The strontium-doped lanthanum cobaltite perovskite was used as a heterogeneous catalyst for the direct oxidative coupling of benzaldehyde with 1,1-diphenylethylene to produce 1,3,3-triphenylprop-2-en-1-one as the major product (scheme 1b). The transformation required the presence of an oxidant, and a quick test indicated that DTBP was appropriate for the reaction system. Initially, reaction conditions were screened to maximize the yield of the coupling product (electronic supplementary material, table S1). The impact of temperature on the formation of the α,β-unsaturated ketone was studied by performing the reaction at different temperatures, ranging from ambient temperature to 140°C (figure 8). The reaction was carried out under air for 18 h, in the presence of 5 mol% perovskite-based catalyst, with four equivalents of DTBP as oxidant. In this protocol, excess benzaldehyde served as both reactant and solvent for the reaction, and no additional organic solvent was required. It was noted that no trace amount of the coupling product was detected for the reaction conducted at room temperature. Boosting the temperature significantly accelerated the coupling reaction, and best yield of 1,3,3-triphenylprop-2-en-1-one was obtained for the reaction performed at 120°C. Nevertheless, considerably lower yield was recorded when the reaction temperature was increased to 140°C. Indeed, GC-MS analysis of the reaction

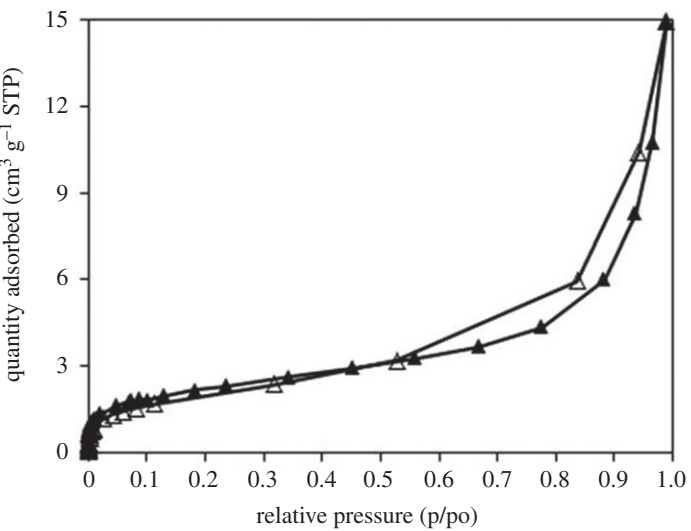

**Figure 3.** Nitrogen adsorption/desorption isotherm of the $La_{0.6}Sr_{0.4}CoO_3$ perovskite. Adsorption data are shown as closed triangles and desorption data as open triangles.

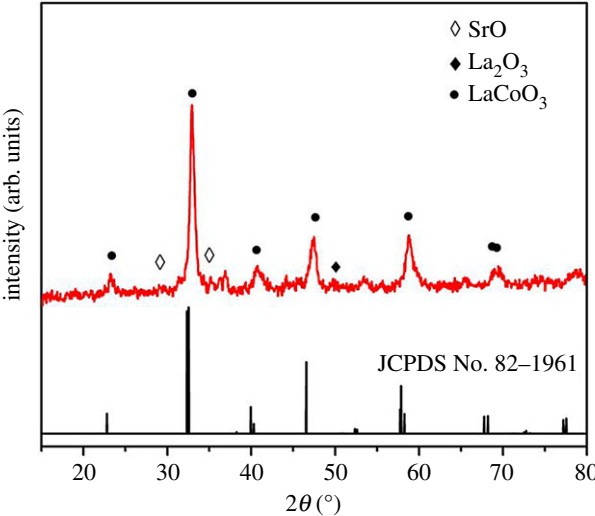

**Figure 4.** XRD patterns of the $La_{0.6}Sr_{0.4}CoO_3$ perovskite catalyst.

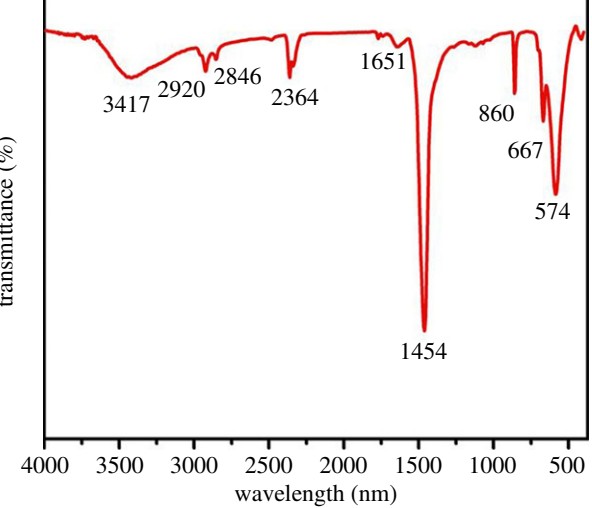

**Figure 5.** FT-IR spectra of the $La_{0.6}Sr_{0.4}CoO_3$ perovskite catalyst.

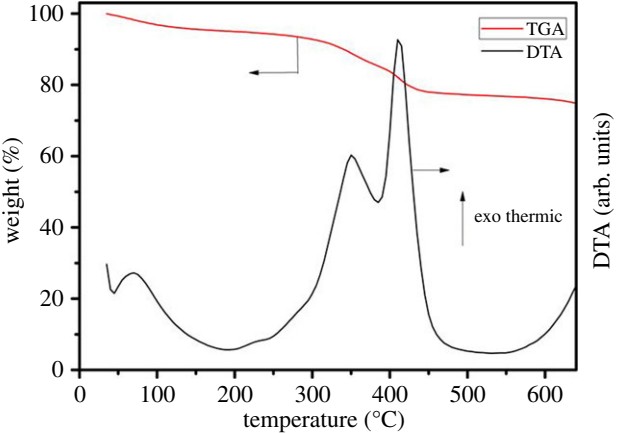

**Figure 6.** TGA of the $La_{0.6}Sr_{0.4}CoO_3$ perovskite catalyst.

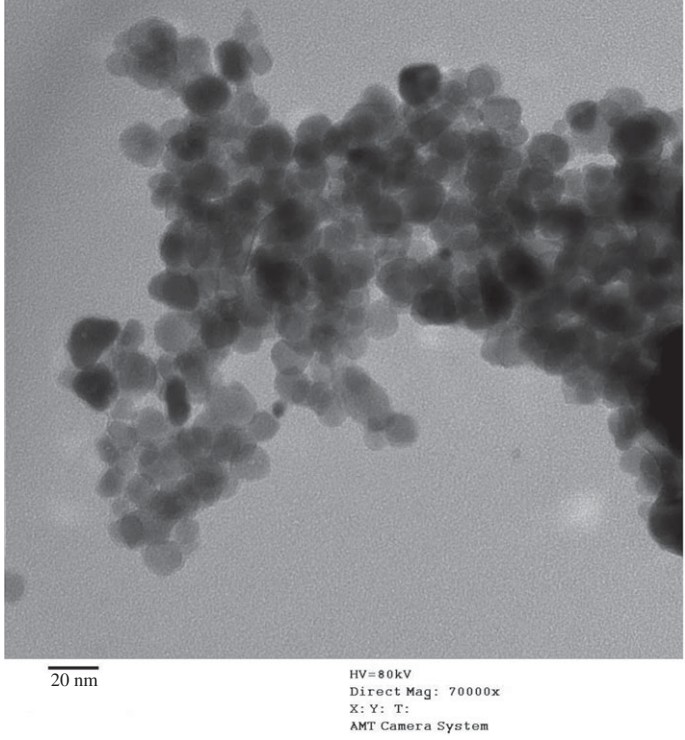

**Figure 7.** TEM micrograph of the $La_{0.6}Sr_{0.4}CoO_3$ perovskite catalyst.

mixture indicated the presence a large amount of benzophenone, implying that the oxidation of 1,1-diphenylethylene was significant under this condition.

It was noted that the coupling reaction could not progress in the absence of the oxidant, with only 7% yield of 1,3,3-triphenylprop-2-en-1-one being detected. A series of oxidants were tested for the reaction in order to maximize the yield of the α,β-unsaturated ketone (figure 9). The reaction using $K_2S_2O_8$ as oxidant afforded 60% yield, while $AgNO_3$, $H_2O_2$ and oxygen were not suitable for this transformation. Indeed, in the previous homogeneous copper-catalysed direct oxidative coupling of alkenes with aldehydes to produce α,β-unsaturated ketones, Wang *et al.* employed *tert*-butyl hydroperoxide as the oxidant [16]. However, in this work, it was observed that the reaction using the strontium-doped lanthanum cobaltite perovskite proceeded slowly in the presence of this oxidant, with 20% and 10% yields being recorded for the case of *tert*-butyl hydroperoxide in water and *tert*-butyl hydroperoxide in decane, respectively. *tert*-Butyl peroxybenzoate and cumyl hydroperoxide also displayed low performance as the oxidant. In this oxidant series, DTBP emerged as the best option, producing the desired α,β-unsaturated ketone in 80% yield. Next, the influence of oxidant amount on the reaction was studied (figure 10). It should be

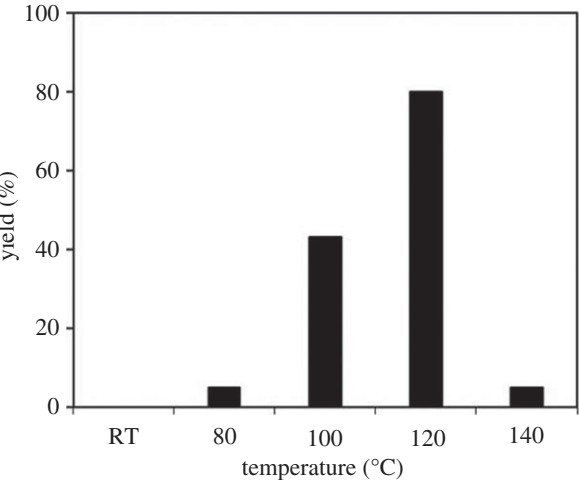

**Figure 8.** Yields of 1,3,3-triphenylprop-2-en-1-one at different temperatures. Reaction conditions: 1,1-diphenylethylene (0.3 mmol); benzaldehyde (1 ml); DTBP (1.2 mmol); catalyst (5 mol%); 18 h; under air. Best yield was achieved at 120°C.

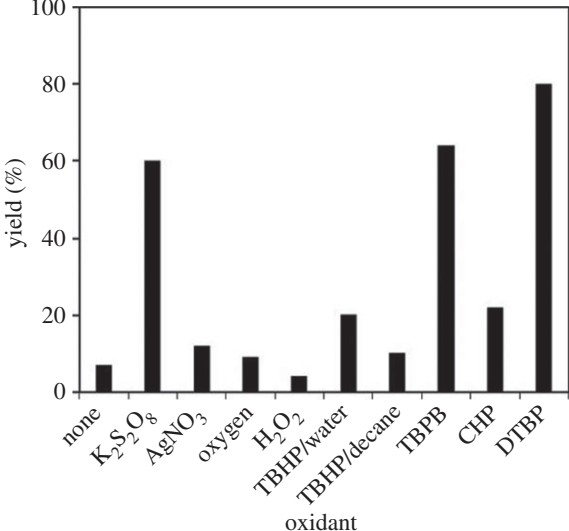

**Figure 9.** Yields of 1,3,3-triphenylprop-2-en-1-one with different oxidants. Reaction conditions: 1,1-diphenylethylene (0.3 mmol); benzaldehyde (1 ml); oxidant (1.2 mmol); catalyst (5 mol%); 120°C; 18 h; under air. Best yield was achieved with DTBP.

noted that only 5% yield was detected in the absence of the oxidant. Expanding the amount of the oxidant led to higher yields, and best yield was achieved with four equivalents of DTBP.

From these results, the amount of the strontium-doped lanthanum cobaltite perovskite catalyst was consequently adjusted to improve the yield of the desired α,β-unsaturated ketone product (figure 11). In the previous homogeneous copper-catalysed direct oxidative coupling of alkenes with aldehydes to produce α,β-unsaturated ketones, Wang *et al.* used up to 20 mol% $CuCl_2$ catalyst for the reaction [16]. Without the catalyst, no trace quantity of product was recorded, verifying the indispensability of the perovskite-based material for the reaction. Using 1 mol% catalyst, 17% yield was detected, and this value was improved to 55% with 3 mol% catalyst. The yield was enhanced to 80% when increasing the catalyst amount to 5 mol%. Nevertheless, extending the catalyst amount to more than 5 mol% did not lead to a higher yield of 1,3,3-triphenylprop-2-en-1-one. Additionally, in this procedure, excess benzaldehyde was used as both reactant and solvent for the reaction. We also explored the coupling reaction in different solvents. For reactions using solid catalysts, the polarity, as well as the structure of solvents, might considerably control the yield and selectivity of the reaction [56]. A series of solvents were tested for the reaction using the strontium-doped lanthanum cobaltite perovskite catalyst under standard conditions (figure 12). Interestingly, it was found that using an additional solvent significantly decelerated the transformation, and the best result was achieved with neat conditions.

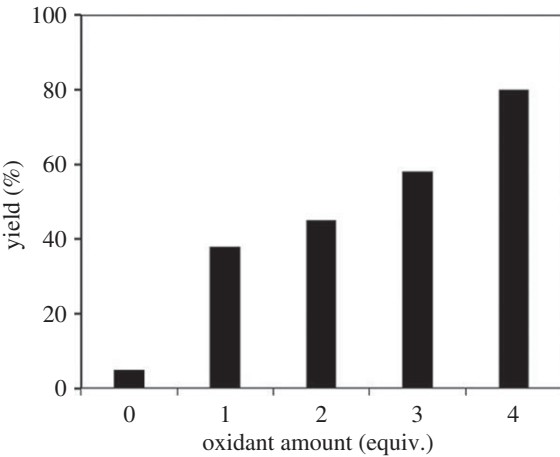

**Figure 10.** Yields of 1,3,3-triphenylprop-2-en-1-one at different oxidant amounts. Reaction conditions: 1,1-diphenylethylene (0.3 mmol); benzaldehyde (1 ml); catalyst (5 mol%); 120°C; 18 h; under air. Best yield was achieved with four equivalents of DTBP.

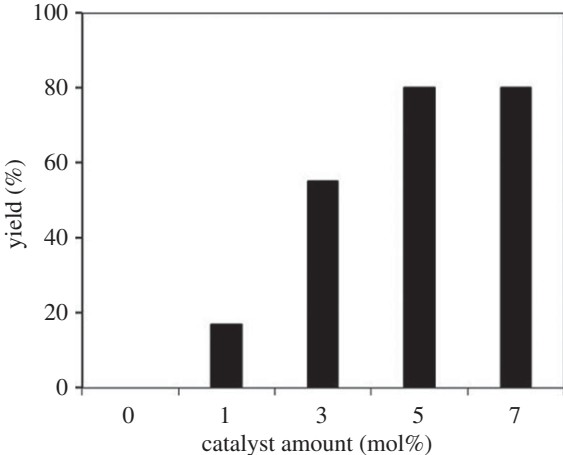

**Figure 11.** Yields of 1,3,3-triphenylprop-2-en-1-one at different catalyst amounts. Reaction conditions: 1,1-DiphenylEthylene (0.3 mmol); benzaldehyde (1 ml); DTBP (1.2 mmol); 120°C; 18 h; under air. Best yield was achieved with 5 mol% catalyst.

Since the direct oxidative coupling of benzaldehyde with 1,1-diphenylethylene to produce 1,3,3-triphenylprop-2-en-1-one using the strontium-doped lanthanum cobaltite perovskite catalyst progressed in a liquid phase, a vital point that must be studied is whether the α,β-unsaturated ketone product was generated via truly heterogeneous or to some extent homogeneous conditions. The reaction was carried out at 120°C under air for 18 h, in the presence of 5 mol% perovskite-based catalyst, with four equivalents of DTBP as oxidant. After the first 3 h with 60% yield being recorded, the solid catalyst was isolated by centrifugation, the liquid phase was obtained and transferred to a new and fresh reactor. The mixture was subsequently heated at 120°C under air for 15 h, with samples being collected and analysed by GC. Under these conditions, almost no additional 1,3,3-triphenylprop-2-en-1-one was observed in the reaction mixture (figure 13). These observations confirmed that the direct oxidative coupling of benzaldehyde with 1,1-diphenylethylene using the strontium-doped lanthanum cobaltite perovskite catalyst proceeded via truly heterogeneous conditions, and almost no α,β-unsaturated ketone was produced via homogeneous catalysis.

In order to highlight the superiority of the strontium-doped lanthanum cobaltite perovskite catalyst in the direct oxidative coupling of benzaldehyde with 1,1-diphenylethylene, numerous homogeneous and heterogeneous catalysts were tested (electronic supplementary material, table S2). The reaction was conducted at 120°C under air for 18 h, in the presence of 5 mol% catalyst, with four equivalents of DTBP as oxidant. In iron salt series, using $FeCl_2$ and $FeCl_3$ led to 38% and 43% yields, respectively, while $Fe(NO_3)_3$ and $FeSO_4$ were inactive for the transformation. Similarly, $CuSO_4$, $Cu(NO_3)_2$ and $Ni(NO_3)_2$ exhibited low activity. The three precursors of the strontium-doped lanthanum cobaltite perovskite, $La(NO_3)_2$, $Co(NO_3)_2$ and $Sr(NO_3)_2$, offered low performance, with 1%, 24% and 1% yields being recorded, respective (figure 14). Moving to heterogeneous catalyst series, the reaction using Cu-MOF-74

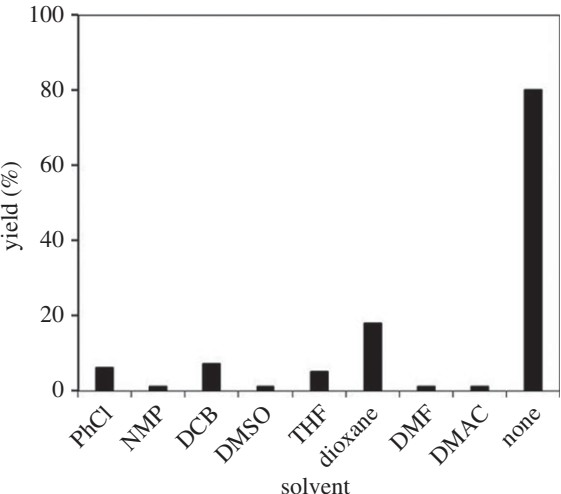

**Figure 12.** Yields of 1,3,3-triphenylprop-2-en-1-one in different solvents. Reaction conditions: 1,1-diphenylethylene (0.3 mmol); benzaldehyde (1 ml); DTBP (1.2 mmol); catalyst (5 mol%); 120°C; 18 h; under air. Best yield was achieved under neat conditions.

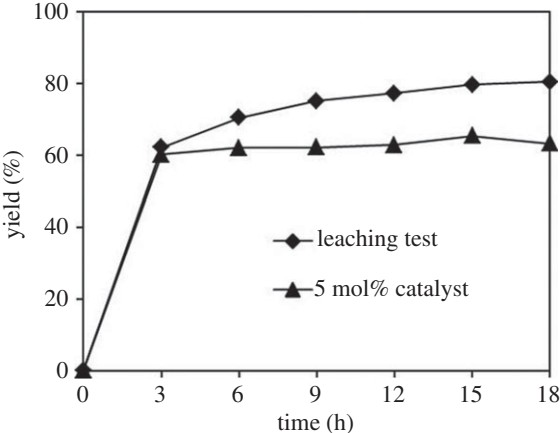

**Figure 13.** Leaching test confirmed that no additional product was produced after catalyst removal.

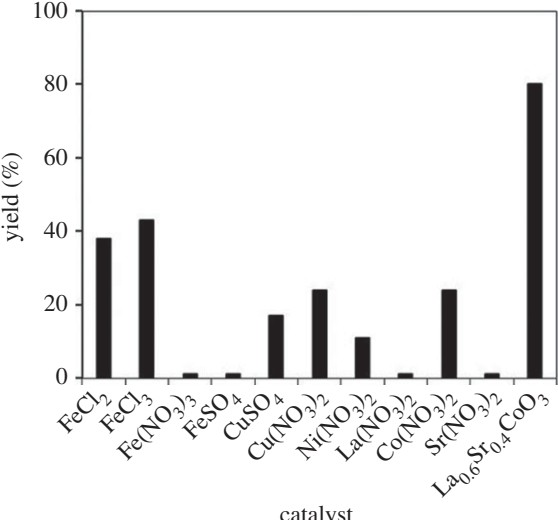

**Figure 14.** Yields of 1,3,3-triphenylprop-2-en-1-one with different homogeneous catalysts. Reaction conditions: 1,1-diphenylethylene (0.3 mmol); benzaldehyde (1 ml); DTBP (1.2 mmol); catalyst (5 mol%); 120°C; 18 h; under air.

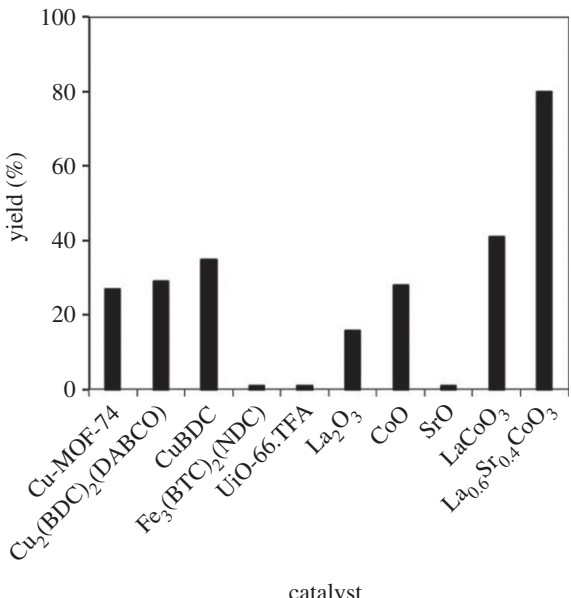

**Figure 15.** Yields of 1,3,3-triphenylprop-2-en-1-one with different heterogeneous catalysts. Reaction conditions: 1,1-diphenylethylene (0.3 mmol); benzaldehyde (1 ml); DTBP (1.2 mmol); catalyst (5 mol%); 120°C; 18 h; under air.

**Scheme 2.** Proposed reaction pathway.

resulted in 27% yield, while 29% and 35% yields were observed for the case of $Cu_2(BDC)_2(DABCO)$ and $Cu(BDC)$, respectively. Other MOFs, including $(BTC)_2(NDC)$, and UiO-66.TFA were inactive for the reaction. Similarly, the $La_2O_3$-catalysed reaction produced 1,3,3-triphenylprop-2-en-1-one in 16% yield, while 28% yield was detected for the case of CoO. SrO was noted to be inactive, with only 1% yield being recorded. Using the undoped perovskite, $LaCoCo_3$, led to 41% yield. Compared to other catalysts, the strontium-doped lanthanum cobaltite perovskite was the best option, offering the desired α,β-unsaturated ketone product in 80% yield (figure 15).

Based on previous reports [16,57–59], a plausible mechanism for the $La_{0.6}Sr_{0.4}CoO_3$-catalysed direct oxidative coupling of benzaldehyde with 1,1-diphenylethylene is proposed in scheme 2. Initially, high-temperature decomposition of *tert*-butyl hydroperoxide with the assistance of Co(II) gives a *tert*-butoxy radical, which abstracts a hydrogen atom of benzaldehyde to form a benzoyl radical. Although such an intermediate was proposed in earlier examples, attempts to trap the benzoyl radical have not been successful [16,57]. Notably, using a radical quenching competitor such as TEMPO lowered the yield, thus somewhat confirming the radical mechanism. Addition of the benzoyl radical to ethene-1,1-diyldibenzene generates a new radical, subsequently followed by the direct oxidation of this radical by

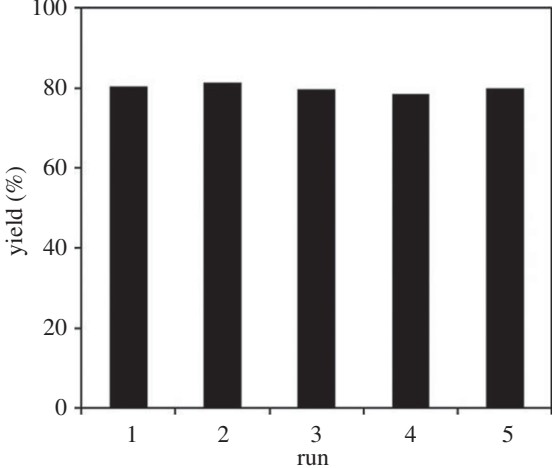

**Figure 16.** Reutilization of the strontium-doped lanthanum cobaltite perovskite catalyst.

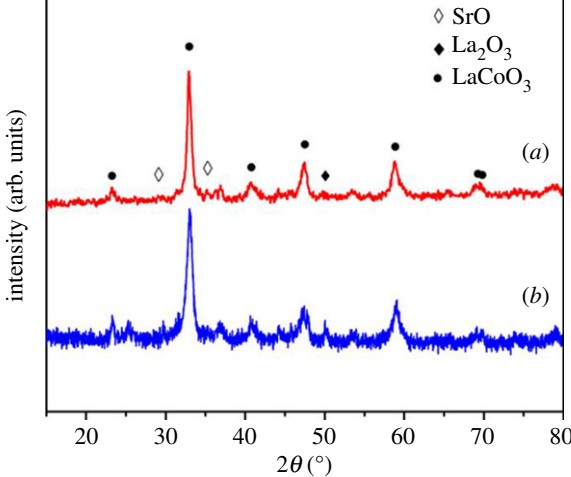

**Figure 17.** XRD patterns of fresh (*a*) and re-used (*b*) $La_{0.6}Sr_{0.4}CoO_3$ perovskite catalysts.

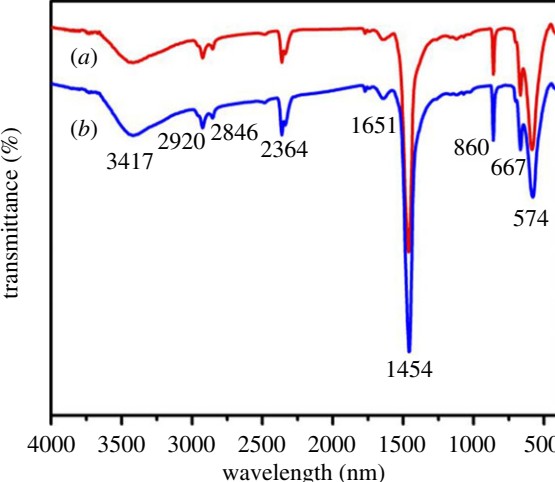

**Figure 18.** FT-IR spectra of fresh (*a*) and re-used (*b*) $La_{0.6}Sr_{0.4}CoO_3$ perovskite catalysts.

Co(III) and the release of a proton, producing the final coupling product, and regenerating the Co(II) species back to the catalytic cycle. It should be noted the reaction using CoO catalyst afforded 28% yield, while 41% yield was detected for that using $LaCoO_3$ catalyst. Indeed, it was previously reported that B site ion in $ABO_3$

**Table 1.** Synthesis of α,β-unsaturated ketones via direct oxidative coupling reactions between 1,1-diphenylethylene and several reactants using the strontium-doped lanthanum cobaltite perovskite catalyst[a]

| entry | reactant 1 | product | yield[b] (%) |
|---|---|---|---|
| 1 | | | 75 |
| 2 | | | 76 |
| 3 | | | 76 |
| 4 | | | 71 |
| 5 | | | 73 |
| 6 | | | 57 |
| 7 | | | 53 |

(*Continued.*)

| entry | reactant 1 | product | yield[b] (%) |
|---|---|---|---|
| 8 | | | 51 |
| 9 | | | 74 |
| 10 | | | 69 |
| 11 | | | 66 |

[a]Reaction conditions: reactant (1 ml); 1,1-diphenylethylene (0.3 mmol); DTBP (1.2 mmol); catalyst (5 mol%); 120°C; 18 h; under air.
[b]Isolated yield.

perovskite with high valence would improve the mobility of lattice oxygen, thus enhancing redox activity of catalyst [60]. Moreover, partial substituting La(III) site by Sr(II) site in the $LaCoO_3$ perovskite would additionally increase the mobility of lattice oxygen to attain the charge balance, further facilitating the redox step between Co(III) and Co(II) species in the catalytic cycle [23]. Therefore, the presence of strontium in the lanthanum cobaltite perovskite would considerably increase the catalytic activity in the direct oxidative coupling reaction. As a result, the $La_{0.6}Sr_{0.4}CoO_3$ exhibited significantly higher catalytic activity than the $LaCoO_3$ as well as the CoO catalyst.

To further underline the strontium-doped lanthanum cobaltite perovskite catalyst in the direct oxidative coupling of benzaldehyde with 1,1-diphenylethylene, a critical issue that must be investigated is the reusability of the catalyst. It was accordingly decided to explore the activity of the catalyst after repeated runs under standard conditions. The reaction was conducted at 120°C under air for 18 h, in the presence of 5 mol% perovskite-based catalyst, with four equivalents of DTBP as oxidant. After the first catalytic experiment, the strontium-doped lanthanum cobaltite perovskite catalyst was separated from the reaction mixture by centrifugation, washed carefully with ethylacetate, acetone, and diethyl ether, dried under vacuum at room temperature overnight on a Schlenkline, and re-used to investigate the possibility of catalyst recycling. It was noted that under standard conditions, the catalyst could be re-used several times for the direct oxidative coupling of benzaldehyde with 1,1-diphenylethylene to produce 1,3,3-triphenylprop-2-en-1-one. Certainly, 80% yield of the desired α,β-unsaturated ketone product was achieved in the fifth catalytic run (figure 16). The recycled catalyst was additionally characterized by XRD, and the result was compared with that of the fresh material. The analysis data verified that the structure of the strontium-doped lanthanum cobaltite perovskite was maintained during the catalytic experiments (figure 17). Additionally, FT-IR spectra of the recycled catalyst were similar to those of the fresh one (figure 18).

With these results, we subsequently expanded the research scope to the direct oxidative coupling reactions between 1,1-diphenylethylene and several reactants using the strontium-doped lanthanum cobaltite perovskite catalyst (table 1). The reaction was conducted at 120°C under air for 18 h, in the

presence of 5 mol% perovskite-based catalyst, with four equivalents of DTBP as oxidant. The α,β-unsaturated ketone product was purified by column chromatography. Using this procedure, 1,3,3-triphenylprop-2-en-1-one was produced in 75% isolated yield (entry 1). Benzaldehydes containing a substituent were also reactive, affording the corresponding product in reasonable yields (entries 2–3). Additionally, heteroaromatic aldehydes such as thiophene-2-carbaldehyde and furan-2-carbaldehyde joined the reaction readily (entries 4–5). Interestingly, we found that several compounds could be used as alternative reactants for this transformation. The reaction of 1,1-diphenylethylene with benzyl alcohol or 4-methylbenzyl alcohol proceeded to 57% and 53% yields, respectively (entries 6–7). Moving to dibenzyl ether, the transformation afforded 1,3,3-triphenylprop-2-en-1-one in 51% yield (entry 8). 2-Oxo-2-phenylacetaldehyde was noted to be less reactive towards the reaction, producing the α,β-unsaturated ketone product in 74% yield (entry 9). Interestingly, (dimethoxymethyl)benzene and 2-bromo-1-phenylethan-1-one were also reactive, with 69% and 66% yields being recorded, respectively (entries 10–11).

# 4. Conclusion

In this work, a strontium-doped lanthanum cobaltite perovskite material was prepared via gelation and calcination approach, and consequently characterized by several methods. The perovskite was used as a recyclable heterogeneous catalyst for the direct oxidative coupling of alkenes with aromatic aldehydes to produce α,β-unsaturated ketones. High yields were achieved in the presence of DTBP as oxidant. The catalyst was re-used several times without considerable decline in the formation of the α,β-unsaturated ketones. Single oxides or salts of strontium, lanthanum and cobalt, and the undoped perovskite exhibited lower catalytic activity than the strontium-doped perovskite. Interestingly, we found that benzaldehyde could be replaced by benzyl alcohol, dibenzyl ether, benzyl amine, 2-bromoacetophenone or (dimethoxymethyl)benzene in the oxidative coupling reaction with alkenes. Reactions between these alternative starting materials with alkenes were not previously reported. The diversified starting materials of the protocol, in conjunction with a recyclable catalyst, make the synthetic protocol efficient and attractive to the chemical and pharmaceutical industries.

Data accessibility. The datasets supporting this article have been uploaded as part of the electronic supplementary material.
Authors' contributions. K.H.T., S.H.D., T.V.H. and P.H.T. designed and performed the catalytic studies. K.H.T. and S.H.D. collected the characterization of products. D.N.P. designed and performed the experiments of material synthesis. M.-V.L. collected and analysed the data of material characterization. T.T.N. and N.T.S.P. prepared the manuscript. All authors gave final approval for publication.
Competing interests. The authors declare no competing interests.
Funding. The Vietnam National University—Ho Chi Minh City (VNU-HCM) is acknowledged for financial support via project no. NCM2019-20-01.
Acknowledgement. We are grateful to Dr Tran Anh Vy (Gachon University, Korea) for TEM analysis.

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
