## [Reviewer comments · Royal Society Open Science]

Review History

RSOS-191313.R0 (Original submission)

Review form: Reviewer 1

Is the manuscript scientifically sound in its present form?

Yes

Are the interpretations and conclusions justified by the results?

Yes

Is the language acceptable?

Yes

Do you have any ethical concerns with this paper?

No

Have you any concerns about statistical analyses in this paper?

Yes

Recommendation?

Major revision is needed (please make suggestions in comments)

Comments to the Author(s)

The authors prepared a catalyst of Sr doped LaCoO₃ for the application of the direct oxidative coupling of alkenes with aromatic aldehydes to produce α,β -unsaturated ketones. The study is novel and it's a valuable report and should be published after a revise. Some comments are listed below.

1. The characterization of physical properties should be provided, such as TEM, XPS and so on, to further prove the type of the material.
2. In Fig. 2, the standard crystal information should be added.
3. In Fig. 5, we can see that the stability of the catalyst is good, and the information has been implied in the FI-IR (Fig. 3) and XRD of Fig.2b. My suggestion is that some of the relevant figures should be put together for a better understanding of the article.
4. In Table 1 and Table 2, the author used different reaction conditions and catalysts to illustrate the effect on conversion rates, respectively. The data is rich and reliable, but wouldn't it be more intuitive to represent such a large amount of data in the tables in some simple figures?
5. In the report, the author proposed the mechanism which as shown in Scheme 2 and in the literature (23) has been proved that Co can act as an active site in the reaction process of CO oxidation by H₂-TPR. So, how do you prove that the reaction of direct oxidative coupling of benzaldehyde with 1, 1-diphenylethylene was going to follow the mechanism which proposed in Scheme 2?
6. In Table 1 Entry 5, the yield has decreased significantly, the explanation is that 'the decomposition of the peroxide at a high temperature'. How to prove it?

Review form: Reviewer 2

Is the manuscript scientifically sound in its present form?

Yes

Are the interpretations and conclusions justified by the results?

Yes

Is the language acceptable?

Yes

Do you have any ethical concerns with this paper?

No

Have you any concerns about statistical analyses in this paper?

No

Recommendation?

Accept with minor revision (please list in comments)

Comments to the Author(s)

This work reports a study on the catalytic behavior of Sr-doped LaCoO₃ perovskite for direct oxidative coupling between alkenes and aldehydes. The authors have demonstrated the synthesis of the perovskite structures by the gelation and calcination method. The obtained perovskite materials have shown good reactivity and durability in catalyzing the cross-coupling of benzaldehyde with 1,1-Diphenylethylene, where an 80% yield of the desired alpha, beta-unsaturated ketone was achieved up to the fifth catalytic run; moreover, the perovskite catalyst was found to be active for a series of aromatic aldehydes. The authors have further discussed the possible reaction mechanism and stated that Sr doping enhanced the reactivity of the perovskite by facilitating redox step in the catalytic cycle. Overall, this research has provided new insights into the understanding of the catalytic behavior of perovskite for organic synthesis and presented a meaningful contribution to advance the research on direct oxidative coupling between alkenes and aldehydes. I suggest that the manuscript be published in Royal Soc. Open Sci. after the authors consider the following minor revisions:

1. The authors have determined element concentration by EDX. This is not sufficient because EDX provides only local element composition while reactivity is an average behavior of the whole catalyst loaded in a reaction batch. Thus, it is necessary to incorporate the stoichiometric ratios (particularly, the doped Sr concentration) of the bulk samples. ICP analysis may be useful for this purpose.
2. The concept of "synergistic effect" (line 38, page 18) is confusing. Further discussion is necessary to interpret it.
3. In order for readers to follow the work, it would be helpful incorporate a summary of the results in the caption of each figure. The authors may also consider making the trend of the catalysts behavior more concise by rationalizing the reaction results (Tables 1 and 2) in term of plot.

Decision letter (RSOS-191313.R0)

02-Sep-2019

Dear Dr Phan:

Title: Alternative pathways to alpha,beta-unsaturated ketones via direct oxidative coupling transformation using Sr-doped LaCoO₃ perovskite catalyst
Manuscript ID: RSOS-191313

The editor assigned to your manuscript has now received comments from reviewers. We would like you to revise your paper in accordance with the referee and Subject Editor suggestions which can be found below (not including confidential reports to the Editor). Please note this decision does not guarantee eventual acceptance.

Please submit your revised paper before 25-Sep-2019. Please note that the revision deadline will expire at 00.00am on this date. If we do not hear from you within this time then it will be assumed that the paper has been withdrawn. In exceptional circumstances, extensions may be possible if agreed with the Editorial Office in advance. We do not allow multiple rounds of revision so we urge you to make every effort to fully address all of the comments at this stage. If

deemed necessary by the Editors, your manuscript will be sent back to one or more of the original reviewers for assessment. If the original reviewers are not available we may invite new reviewers.

Please also include the following statements alongside the other end statements. As we cannot publish your manuscript without these end statements included, if you feel that a given heading is not relevant to your paper, please nevertheless include the heading and explicitly state that it is not relevant to your work.

- Acknowledgements

RSC Associate Editor:
Comments to the Author:
(There are no comments.)

RSC Subject Editor:
Comments to the Author:
(There are no comments.)

Reviewers' Comments to Author:

Reviewer: 1

Comments to the Author(s)

The authors prepared a catalyst of Sr doped LaCoO₃ for the application of the direct oxidative coupling of alkenes with aromatic aldehydes to produce α,β -unsaturated ketones. The study is novel and it's a valuable report and should be published after a revise. Some comments are listed below.

1. The characterization of physical properties should be provided, such as TEM, XPS and so on, to further prove the type of the material.
2. In Fig. 2, the standard crystal information should be added.
3. In Fig. 5, we can see that the stability of the catalyst is good, and the information has been implied in the FT-IR (Fig. 3) and XRD of Fig.2b. My suggestion is that some of the relevant figures should be put together for a better understanding of the article.
4. In Table 1 and Table 2, the author used different reaction conditions and catalysts to illustrate the effect on conversion rates, respectively. The data is rich and reliable, but wouldn't it be more intuitive to represent such a large amount of data in the tables in some simple figures?
5. In the report, the author proposed the mechanism which as shown in Scheme 2 and in the literature (23) has been proved that Co can act as an active site in the reaction process of CO oxidation by H₂-TPR. So, how do you prove that the reaction of direct oxidative coupling of benzaldehyde with 1, 1-diphenylethylene was going to follow the mechanism which proposed in Scheme 2?
6. In Table 1 Entry 5, the yield has decreased significantly, the explanation is that 'the decomposition of the peroxide at a high temperature'. How to prove it?

Reviewer: 2

Comments to the Author(s)

This work reports a study on the catalytic behavior of Sr-doped LaCoO₃ perovskite for direct oxidative coupling between alkenes and aldehydes. The authors have demonstrated the synthesis of the perovskite structures by the gelation and calcination method. The obtained perovskite materials have shown good reactivity and durability in catalyzing the cross-coupling of benzaldehyde with 1,1-Diphenylethylene, where an 80% yield of the desired α,β -unsaturated ketone was achieved up to the fifth catalytic run; moreover, the perovskite catalyst was found to be active for a series of aromatic aldehydes. The authors have further discussed the possible reaction mechanism and stated that Sr doping enhanced the reactivity of the perovskite by facilitating redox step in the catalytic cycle. Overall, this research has provided new insights into the understanding of the catalytic behavior of perovskite for organic synthesis and presented a meaningful contribution to advance the research on direct oxidative coupling between alkenes and aldehydes. I suggest that the manuscript be published in Royal Soc. Open Sci. after the authors consider the following minor revisions:

1. The authors have determined element concentration by EDX. This is not sufficient because EDX provides only local element composition while reactivity is an average behavior of the whole catalyst loaded in a reaction batch. Thus, it is necessary to incorporate the stoichiometric ratios (particularly, the doped Sr concentration) of the bulk samples. ICP analysis may be useful for this purpose.
2. The concept of "synergistic effect" (line 38, page 18) is confusing. Further discussion is necessary to interpret it.
3. In order for readers to follow the work, it would be helpful incorporate a summary of the results in the caption of each figure. The authors may also consider making the trend of the catalysts behavior more concise by rationalizing the reaction results (Tables 1 and 2) in term of plot.

Author's Response to Decision Letter for (RSOS-191313.R0)

See Appendix A.

RSOS-191313.R1 (Revision)

Review form: Reviewer 1

Is the manuscript scientifically sound in its present form?

Yes

Are the interpretations and conclusions justified by the results?

Yes

Is the language acceptable?

Yes

Do you have any ethical concerns with this paper?

No

Have you any concerns about statistical analyses in this paper?

Yes

Recommendation?

Accept as is

Comments to the Author(s)

The authors have addressed what I am concerned

Review form: Reviewer 2

Is the manuscript scientifically sound in its present form?

Yes

Are the interpretations and conclusions justified by the results?

Yes

Is the language acceptable?

Yes

Do you have any ethical concerns with this paper?

No

Have you any concerns about statistical analyses in this paper?

No

Recommendation?

Accept as is

Comments to the Author(s)

The authors have addressed my concerns and the manuscript is ready for publication.

Decision letter (RSOS-191313.R1)

28-Oct-2019

Dear Dr Phan:

Title: Alternative pathways to alpha,beta-unsaturated ketones via direct oxidative coupling transformation using Sr-doped LaCoO₃ perovskite catalyst
Manuscript ID: RSOS-191313.R1

It is a pleasure to accept your manuscript in its current form for publication in Royal Society Open Science. The chemistry content of Royal Society Open Science is published in collaboration with the Royal Society of Chemistry.

RSC Associate Editor:
Comments to the Author:
(There are no comments.)

RSC Subject Editor:
Comments to the Author:
(There are no comments.)

Reviewer(s)' Comments to Author:
Reviewer: 1

Comments to the Author(s)
The authors have addressed what I am concerned

Reviewer: 2

Comments to the Author(s)
The authors have addressed my concerns and the manuscript is ready for publication.

Appendix A

Title: Alternative pathways to alpha,beta-unsaturated ketones via direct oxidative coupling transformation using Sr-doped LaCoO₃ perovskite catalyst

Manuscript ID: RSOS-191313

Dear Dr Laura Smith,

Thank you very much for your consideration.

I would like to thank the reviewers for their helpful comments on the manuscript. We have modified the manuscript accordingly, and detailed corrections are listed below point by point:

Reviewers' comments:

Reviewer: 1

Comments to the Author(s)

The authors prepared a catalyst of Sr doped LaCoO₃ for the application of the direct oxidative coupling of alkenes with aromatic aldehydes to produce α,β -unsaturated ketones. The study is novel and it's a valuable report and should be published after a revise. Some comments are listed below.

1. The characterization of physical properties should be provided, such as TEM, XPS and so on, to further prove the type of the material.

- We thank the reviewer for this helpful comment. We definitely agree with the reviewer. More experiments were carried out for this point, and new results have been added to the manuscript to further prove the type of the material.
- Nitrogen adsorption/desorption isotherm result has been added as Fig. 3:

“Fig. 3. Nitrogen adsorption/desorption isotherm of the $\text{La}_{0.6}\text{Sr}_{0.4}\text{CoO}_3$ perovskite. Adsorption data are shown as closed triangles and desorption data as open triangles.”

- TGA result has been added as Fig. 6:

“Fig. 6. TGA of the $\text{La}_{0.6}\text{Sr}_{0.4}\text{CoO}_3$ perovskite catalyst”

- TEM result has been added as Fig. 7:

“Fig. 7. TEM micrograph of the $\text{La}_{0.6}\text{Sr}_{0.4}\text{CoO}_3$ perovskite catalyst.”

- ICP analysis result has been added to confirm the composition of the catalyst: “Additionally, ICP analysis of the bulk sample indicated a La:Sr:Co molar ratio of 0.63:0.38:1.0.”
- Unfortunately, we cannot run XPS of the catalyst as there is no XPS system in our country. Nevertheless, we believe that material can be confirmed by XRD, FT-IR, nitrogen

adsorption/desorption isotherm, SEM, EDX, TEM, TGA, ICP. We have carefully checked previous works on perovskite, and many perovskite, including doped LaCoO_3 , can be confirmed without XPS.

2. *In Fig. 2, the standard crystal information should be added.*

- We thank the reviewer for this helpful comment. We definitely agree with the reviewer.
- The standard crystal information has been added to the XRD result of the material in Fig. 4.

3. *In Fig. 5, we can see that the stability of the catalyst is good, and the information has been implied in the FI-IR (Fig. 3) and XRD of Fig.2b. My suggestion is that some of the relevant figures should be put together for a better understanding of the article.*

- We thank the reviewer for this helpful comment. We definitely agree with the reviewer.
- As suggested, the FT-IR and XRD of the reused catalyst have been put together with the catalyst recycling studies for a better understanding of the article (Fig. 16, Fig. 17 and Fig. 18).

4. *In Table 1 and Table 2, the author used different reaction conditions and catalysts to illustrate the effect on conversion rates, respectively. The data is rich and reliable, but wouldn't it be more intuitive to represent such a large amount of data in the tables in some simple figures?*

- We thank the reviewer for this helpful comment. We definitely agree with the reviewer.
- As suggested, we have replaced Table 1 and Table 2 by relevant figures (from Fig. 8 to Fig. 15).
- We have moved Table 1 and Table 2 to the Supporting information as Table S1 and Table S2.

5. *In the report, the author proposed the mechanism which as shown in Scheme 2 and in the*

literature (23) has been proved that Co can act as an active site in the reaction process of CO oxidation by H₂-TPR. So, how do you prove that the reaction of direct oxidative coupling of benzaldehyde with 1, 1-diphenylethylene was going to follow the mechanism which proposed in Scheme 2?

- We thank the reviewer for this helpful comment. We definitely agree with the reviewer. More experiments were carried out for this point.
- We attempted to trap the benzoyl radical which should be involved in the mechanism as we proposed. Increasing the amount of a radical quencher such as TEMPO did lower the reaction yield. Unfortunately, no adduct was isolable or observed. Although a previous report of Li and co-workers presented that such a TEMPO-benzoyl species could be formed,¹ our conditions were much different. We hypothesized that the use of “homogeneous” iron species in Li’s method would prolong the lifetime of the benzoyl radical.
- A short description on our attempt was added in the manuscript as “**Although such an intermediate was proposed in earlier examples, attempts to trap the benzoyl radical have not been successful.^{16,57} Notably, using a radical quenching competitor such as TEMPO lowered the yield, thus somewhat confirming the radical mechanism**”

Reference: 1. W. Liu, Y. Li, K. Liu, and Z. Li, *J. Am. Chem. Soc.* **2011**, *133*, 10756.

6. In Table 1 Entry 5, the yield has decreased significantly, the explanation is that ‘the decomposition of the peroxide at a high temperature’. How to prove it?

- We thank the reviewer for this helpful comment. We definitely agree with the reviewer. Please allow us to clarify this point.
- More experiments were carried out for this point. The experiment at 140 °C was repeated. We analyzed the reaction mixture by GC-MS. We could not detect the decomposition product of the peroxide. Instead, a large amount of benzophenone was detected by GC-MS, due to the decomposition of 1,1-diphenylethylene.

- We have modified the text to “ Nevertheless, considerably lower yield was recorded when the reaction temperature was increased to 140 °C. Indeed, GC-MS analysis of the reaction mixture indicated the presence a large amount of benzophenone, implying that the oxidation of 1,1-diphenylethylene was significant under this condition.”

Reviewer: 2

Comments to the Author(s)

This work reports a study on the catalytic behavior of Sr-doped LaCoO₃ perovskite for direct oxidative coupling between alkenes and aldehydes. The authors have demonstrated the synthesis of the perovskite structures by the gelation and calcination method. The obtained perovskite materials have shown good reactivity and durability in catalyzing the cross-coupling of benzaldehyde with 1,1-Diphenylethylene, where an 80% yield of the desired alpha, beta-unsaturated ketone was achieved up to the fifth catalytic run; moreover, the perovskite catalyst was found to be active for a series of aromatic aldehydes. The authors have further discussed the possible reaction mechanism and stated that Sr doping enhanced the reactivity of the perovskite by facilitating redox step in the catalytic cycle. Overall, this research has provided new insights into the understanding of the catalytic behavior of perovskite for organic synthesis and presented a meaningful contribution to advance the research on direct oxidative coupling between alkenes and aldehydes. I suggest that the manuscript be published in Royal Soc. Open Sci. after the authors consider the following minor revisions:

1. The authors have determined element concentration by EDX. This is not sufficient because EDX provides only local element composition while reactivity is an average behavior of the whole catalyst loaded in a reaction batch. Thus, it is necessary to incorporate the stoichiometric ratios (particularly, the doped Sr concentration) of the bulk samples. ICP analysis may be useful for this purpose.

- We thank the reviewer for this helpful comment. We definitely agree with the reviewer. More experiments were carried out for this point.
- As suggested, we conducted ICP analysis of the bulk samples of the catalyst, and the result has been added to the manuscript.
- More texts have been added to the manuscript **“Additionally, ICP analysis of the bulk sample indicated a La:Sr:Co molar ratio of 0.63:0.38:1.0.”**

2. The concept of “synergistic effect” (line 38, page 18) is confusing. Further discussion is necessary to interpret it.

- We thank the reviewer for this helpful comment. We definitely agree with the reviewer. Please allow us to clarify this point.
- The words “synergistic effect” might be unclear. Therefore we have modified the texts and further discuss about it.
- The new texts for this point are:
“Indeed, it was previously reported that B site ion in ABO_3 perovskite with high valence would improve the mobility of lattice oxygen, thus enhancing redox activity of catalyst⁶⁰. Moreover, partial substituting La(III) site by Sr(II) site in the $LaCoO_3$ perovskite would additionally increase the mobility of lattice oxygen to attain the charge balance, further facilitating the redox step between Co(III) and Co(II) species in the catalytic cycle²³. Therefore, the presence of strontium in the lanthanum cobaltite perovskite would considerably increase the catalytic activity in the direct oxidative coupling reaction. As a result, the $La_{0.6}Sr_{0.4}CoO_3$ exhibited significantly higher catalytic activity than the $LaCoO_3$ as well as the CoO catalyst.”

3. In order for readers to follow the work, it would be helpful incorporate a summary of the results in the caption of each figure. The authors may also consider making the trend of the catalysts behavior more concise by rationalizing the reaction results (Tables 1 and 2) in term of plot.

- We thank the reviewer for this helpful comment. We definitely agree with the reviewer.
- As suggested, we have replaced Table 1 and Table 2 by relevant figures (from Fig. 8 to Fig. 15).
- We have moved Table 1 and Table 2 to the Supporting information as Table S1 and Table S2.
- As suggested, a summary of the results have been added to the caption of each figure (from Fig. 8 to Fig. 15).

Thank you very much again for your consideration.

I look forwards to hearing from you soon.

Sincerely yours,

Nam Phan